# Discovering Meaningful Units with Visually Grounded Semantics from Image Captions

Melika Behjati*                                                      *melikabehjati@gmail.com*

**James Henderson**                                              *james.henderson@idiap.ch*
*Idiap Research Institute*
*Martigny, Switzerland*

**Reviewed on OpenReview:** *https://openreview.net/forum?id=kndKGnEOtb*

## Abstract

Fine-grained knowledge is crucial for vision-language models to obtain a better understanding of the real world. While there has been work trying to acquire this kind of knowledge in the space of vision and language, it has mostly focused on aligning the image patches with the tokens on the language side. However, image patches do not have any meaning to the human eye, and individual tokens do not necessarily carry groundable information in the image. It is groups of tokens which describe different aspects of the scene. In this work, we propose a model which groups the caption tokens as part of its architecture in order to capture a fine-grained representation of the language. We expect our representations to be at the level of objects present in the image, and therefore align our representations with the output of an image encoder trained to discover objects. We show that by learning to group the tokens, the vision-language model has a better fine-grained understanding of vision and language. In addition, the token groups that our model discovers are highly similar to groundable phrases in text, both qualitatively and quantitatively.

## 1 Introduction

Vision-language models have been shown to be less effective at capturing fine-grained information (e.g., understanding relationships and recognizing verbs) about the images described by the captions (Yuksekgonul et al., 2022; Bugliarello et al., 2023; Kamath et al., 2023; Dumpala et al., 2024; Pearson et al., 2025). This information is crucial for the models to obtain a better understanding of the real world. While there has been work trying to acquire this kind of knowledge in the space of vision and language, it has mostly focused on aligning the image patches with the input tokens on the language side (Yao et al., 2022; Wang et al., 2022; Zeng et al., 2022; Mukhoti et al., 2023; Zhang et al., 2024). However, image patches do not have any meaning to the human eye, and individual tokens often do not carry information groundable in the image, neither do the single-vector representations. Minimally, it is groups of image patches which represent objects and the group of tokens in the text that refer to those objects. For this reason, there has been an active line of research in vision investigating the unsupervised discovery of objects by learning to assign image patches to their representative object slots (Locatello et al., 2020; Wu et al., 2024; Didolkar et al., 2025). Xu et al. (2022) integrated an object discovery module into their vision-language model to learn the object entities. They showed that representing the image at the level of its constituent objects improves the performance of their model in downstream tasks. In this paper, we investigate the unsupervised discovery of groundable phrases on the language side to get better correspondence with objects on the vision side. We hypothesize that finding these meaningful units in language representations will improve the fine-grained understanding of image-caption semantic relationships. As far as we are aware, we are the first to investigate this possibility.

---

*Work done while at Idiap Research Institute and EPFL.

We base our model on the model of visual object discovery using image caption pairs proposed by Xu et al. (2022). We freeze the image side of the model, and introduce analogous deep learning mechanisms to discover *objects*[1] on the language side (see Figure 1). We investigate two types of losses, one which promotes the correspondence between representations of the language side and representations on the vision side, and one which promotes the ability to reconstruct the text from the language representations. We find that training with both these losses leads to better fine-grained understanding of the image-text relationship, and discovers units which are highly similar to groundable phrases in text, both qualitatively and quantitatively. Further analysis finds that optimizing the image-text correspondence alone does not lead to the discovery of meaningful units on the language side, and while this model does learn a good fine-grained understanding of the image-text relationship, it does not represent the semantics of objects as well as the model which does represent groundable phrases. We also find that optimizing the reconstruction loss alone does lead to the discovery of meaningful units on the language side, but they have a slightly worse similarity to groundable phrases than the model which includes grounding information, and do not capture image-text relationships.

Our contributions are as follows,

- We develop a novel model to discover meaningful units from the image captions in the vision language setup (Section 2.1).

- We show that our model has better fine-grained vision and language understanding compared to a single-vector representation of text, under two different benchmarks (Section 4.2).

- We show that the segments that our model discovers are meaningful both qualitatively, and in terms of accordance with human-annotated groundable phrases (Section 4.3).

## 2 Method

To facilitate learning the fine-grained semantics of image-text relationships, we propose a model for learning text representations whose granularity matches the granularity of objects in the image, meaning that it is neither as coarse-grained as having a single vector for embedding the entire text[2] nor as fine-grained as having a different vector for every token. Given a dataset of image-caption pairs, $D = \{(I_i, T_i)\}_{i=1,...,N}$, we want to learn a representation of each caption in the form of groups of tokens which are aligned with the semantic space of objects in its image. To do so, we freeze the image encoder which has been trained to output the objects in the image and only train the text encoder and the projection heads. The input representation of language is at the level of subword tokens, so we aim to find a more abstract representation which would approximately represent groundable phrases, for example as defined by the human annotators of Plummer et al. (2015). More specifically, let $T_i = [t_{i1}, \ldots, t_{iM}]$, where $t_{ij}$ is a token of $T_i$ and $M$ is the total number of tokens in $T_i$. We would like to group the tokens $t_{ij}$s into non-overlapping groups $T_i = \{g_{i1}^T, \ldots, g_{iK}^T\}$ where $K < M$ and every $t_{ij}$ is assigned to a group. This would lead to a more compact abstract representation of $T_i$. In this work we treat the number of groups $K$ as a fixed hyperparameter similar to (Xu et al., 2022; Seitzer et al., 2023; Singh et al., 2022), leaving the identification of the number of groundable phrases in a given example to future work.

### 2.1 Model

We illustrate an overview of our model in Figure 1 and describe each of its components in the following sections.

#### 2.1.1 Text Encoder: Text Group Transformer

We design our text encoder to learn semantic units of language at the level of objects in the image, by grouping the caption tokens. The key idea is to have shared learnable group vectors which can bind to different tokens of input (Xu et al., 2022). At each stage, the groups carry the information from the previous

---

[1]We use the terms *objects*, *groups* and *units* interchangeably.
[2]This is the common way of representing text in dual-stream vision-language models like CLIP (Radford et al., 2021).

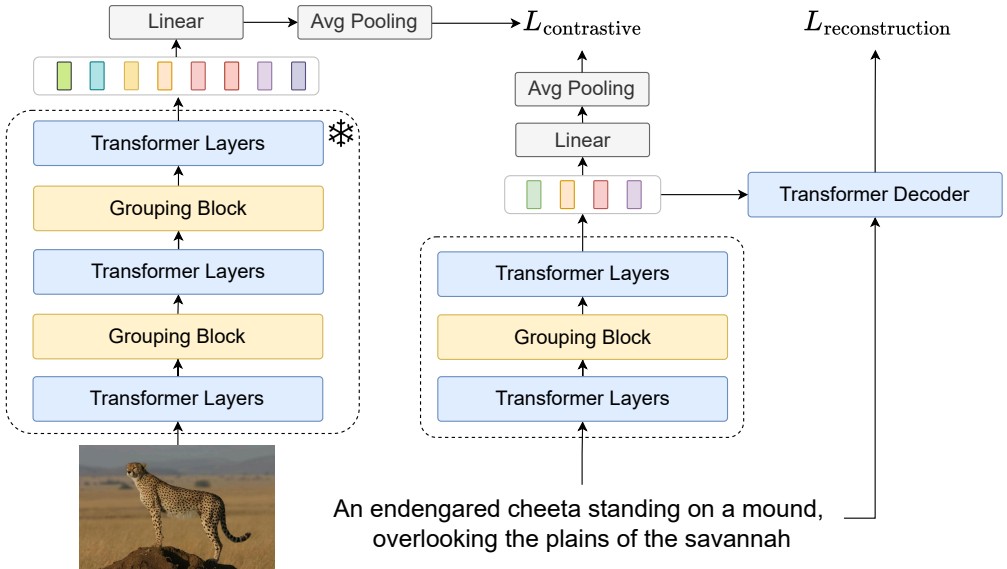

Figure 1: Overview of the model[*]. We freeze the image encoder and only train the text encoder, decoder, and the linear projection heads. The image passes through Transformer layers followed by the grouping blocks. The output of the image encoder is a set of groups which are approximately representing the objects. The caption also passes through the same set of blocks and the output of the text encoder is a set of groups representing units in language. The two modalities interact via a contrastive loss. There is also a reconstruction loss where the decoder decodes the text groups into the original input.

[*] The image and caption are selected from the GCC 3M dataset without any modifications.

layer to the next layer. To initiate the binding, the groups are appended to the input tokens they need to bind, and they all interact via several Transformer encoder layers to allow the groups and tokens to exchange information. Then, by performing a top-down attention mechanism shown as the Grouping block, the groups bind to different parts of the input.

More specifically, we first embed the input tokens and add learned positional encodings to them. Then, we append the learnable group vectors, $[g_{ik}^T]_{k=1\ldots K}$, to these embedded inputs, $[t_{ij}]_{j=1\ldots M}$, and pass the resulting vectors through some Transformer encoder layers, allowing them to interact with each other. We denote the encoded tokens and groups as $\hat{t}_{ij}$ and $\hat{g}_{ik}$. Then the grouping happens in a grouping block. In this block, the groups act as the queries and the encoded inputs as keys and values through a top-down attention mechanism. As with standard attention, the raw attention scores are computed as

$$A_{kj}^{\text{raw}} = \frac{Q(\hat{g}_{ik}^T)K^\intercal(\hat{t}_{ij})}{\sqrt{d}} \tag{1}$$

where $d$ is the dimension of the model and $Q$ and $K$ are linear query and key projections. In order to have discrete assignments of inputs to the groups, GroupViT actually performs a hard assignment over $A^{\text{raw}}$ by utilizing Gumble softmax (Jang et al., 2017; Maddison et al., 2017). Namely,

$$A' = \text{GumbleSoftmax}(A^{\text{raw}}). \tag{2}$$

In top-down attention, instead of normalizing over the keys in the softmax function, the $A'$ weights are first normalized over the queries, which are the groups. This will make the groups compete for representing different inputs (Locatello et al., 2020) and has been shown to be the most important component in discovering objects (Wu et al., 2023). After the normalization, the hard assignment happens and the gradient is backpropagated with the straight through trick (Van Den Oord et al., 2017), that is:

$$A = \text{one-hot}(\text{argmax}_{\text{groups}}(A')) - \text{sg}(A') + A' \tag{3}$$

where sg is the stop gradient operator. Finally, the group vectors get updated as

$$\bar{g}_{ik}^T = \hat{g}_{ik}^T + W(\sum_j \frac{A_{kj}}{\sum_j A_{kj}} V(t_{ij})) \tag{4}$$

where $V$ and $W$ are the linear projections for values and outputs, respectively.

After the grouping block, the updated group vectors serve as inputs to subsequent Transformer encoder layers. Finally, these refined groups represent the fine-grained semantics of the text in our model.

### 2.1.2  Image Encoder

We use the image encoder of Xu et al. (2022), which follows the same architecture as the text encoder, but with two stacked levels of transformer encoder layers and grouping blocks. As its input, the images are first divided into patches and then linearly projected. The encoder then extracts the set of image groups denoted as $\{\bar{g}_{ik}^I\}$. Due to the computational cost, we freeze the image encoder and assume that the image groups are representing objects in the image.

### 2.2  Training Objectives

Our model is trained with two different losses, i.e., a contrastive loss and a reconstruction loss, which we will explain in the following. The two losses are combined with a hyperparameter $\lambda$ which controls the ratio between the two terms.

$$L_{\text{total}} = L_{\text{contrastive}} + \lambda L_{\text{reconstruction}} \tag{5}$$

### 2.2.1  Contrastive Loss

The image and text modalities interact via a contrastive loss. First, the final groups for each modality are mapped into a common space with a Linear projector ($\Phi^T$), i.e., $z_{ij}^T = \Phi^T(\bar{g}_{ij}^T)$. Then, we average pool over them to obtain the global features for each modality ($\hat{z}_i^T$). We compute the InfoNCE loss (Oord et al., 2018) for every modality separately. Given a batch size of $B$, a similarity function (sim) and temperature $\tau$, the infoNCE loss for the image to text is

$$L_{\text{I-T}} = -\frac{1}{B} \sum_{i=1}^{B} \log \frac{e^{\text{sim}(\hat{z}_i^T, \hat{z}_i^I)/\tau}}{\sum_{j=1}^{B} e^{\text{sim}(\hat{z}_j^T, \hat{z}_i^I)/\tau}}, \tag{6}$$

and respectively for the text to image is

$$L_{\text{T-I}} = -\frac{1}{B} \sum_{i=1}^{B} \log \frac{e^{\text{sim}(\hat{z}_i^T, \hat{z}_i^I)/\tau}}{\sum_{j=1}^{B} e^{\text{sim}(\hat{z}_i^T, \hat{z}_j^I)/\tau}}. \tag{7}$$

The final contrastive loss is calculated by averaging the two losses,

$$L_{\text{contrastive}} = \frac{1}{2}(L_{\text{I-T}} + L_{\text{T-I}}). \tag{8}$$

As for the similarity function sim(a,b), we consider the cosine similarity between the vectors.

### 2.2.2  Reconstruction Loss

In order to encourage the model to group the tokens into meaningful units, we incorporate a reconstruction loss from a text decoder. This loss encourages the model to assign tokens to different groups in order to spread information about the text across multiple vectors, and thus make better use of the available vectors.

We employ a simple shallow Transformer decoder to reconstruct the original input conditioned on the text groups. The shallow decoder has to rely on the representation in the groups for decoding, thus putting the burden on the encoder to better encode the information about the input in the groups (Bowman et al., 2016).

The output of this layer is

$$\overline{T}_i = \text{TransformerDecoder}(T_i | \{g_{i1}^T \ldots g_{iK}^T\}). \tag{9}$$

The probabilities from these predictions are then used to define the reconstruction loss:

$$L_{\text{reconstruction}} = \sum_{i=1}^{B} \text{CE}(\overline{T}_i, T_i | \{g_{i1}^T, \ldots, g_{iK}^T)\}) \tag{10}$$

where CE is the cross entropy between the output probabilities of the decoder and the original input given the discovered groups.

## 3 Related Work

Our work is related to different tasks in vision and language, which we will explain in this section.

**Object discovery.** Here the task is to discover the objects in an image or video without any supervision. Slot-based object discovery (Locatello et al., 2020) has become popular due to the simplicity of the method (Singh et al., 2022; Sajjadi et al., 2022; Singh et al., 2023a; Seitzer et al., 2023; Singh et al., 2023b; Wu et al., 2023; 2024; Didolkar et al., 2025). We have a novel adaptation of this method in discovering units similar to phrases in language with visually grounded semantics.

**Weakly supervised visual grounding.** Visual grounding refers to the tasks where a phrase or expression is grounded in the image. In the weakly supervised setup, the only information used is the pairing of the image with its caption (Datta et al., 2019; Gupta et al., 2020; Wang et al., 2020; Chen et al., 2022; He et al., 2024; Kuang et al., 2025). In referring expression comprehension and referring image segmentation, the model must identify a specific part of the image described in a single expression. Kim et al. (2023) addressed the task of referring image segmentation by employing a slot-based object discovery module and merging relevant slots by cross attending over them with the textual query to build the final segmentation. In this task, the phrases are predetermined and no discovery happens on the language side, which is what our model is designed to do.

**Vision language models with vision and language alignments.** While many large-scale vision language models have been developed, it has been shown that they fall short in understanding fine-grained details in the image. This is especially more pronounced in the dual-stream Vision Language Models (VLMs) like CLIP (Radford et al., 2021), where the modalities interact only via a single-vector representation. Therefore, there has been efforts to align language and vision at the level of patches and tokens (Yao et al., 2022; Wang et al., 2022; Mukhoti et al., 2023; Jing et al., 2024; Zhang et al., 2024; Xie et al., 2025). Zeng et al. (2022) uses additional supervision from phrase grounding annotations to help the model learn the alignments. Bica et al. (2024) aligns tokens and patch embeddings at different levels of granularity simultaneously. Li et al. (2022) learns the semantic alignment from the perspective of game-theoretic interactions.

**Object detection.** The objective of this task is to detect the object boundaries in an image with a supervised objective. Our work is related to query-based object detection, such as the approach in (Carion et al., 2020; Kamath et al., 2021), where, at decoding time, learnable object queries attend to the input features and encode an object. Liu et al. (2023) extend this approach by proposing a dual query model, demonstrating that simultaneously learning phrases and their corresponding objects improves the module's groundable understanding. The main difference between our model and this line of work lies in the weakly supervised nature of our approach.

**Zero-shot open-vocabulary semantic segmentation.** Semantic segmentation is a well-established task in computer vision. Recently, with the rise of VLMs, these models have demonstrated promising zero-shot capabilities in the semantic segmentation task as well. (Xu et al., 2022) propose a hierarchical grouping architecture that learns to group image regions without pixel-level annotations, relying solely on paired image and text data. Patel et al. (2023) expanded on image-text alignment, suggesting to not only align an

image to the corresponding text but also to the text from visually similar samples. Additionally, Mukhoti et al. (2023) propose aligning patch tokens from a vision encoder with the <cls> token from a text encoder to enhance the model's performance.

**Unit discovery in language.** Lately, discovering language units as part of the model architecture has been explored. These models operate on top of characters, where the units are usually at the level of subwords or words. The purpose is to optimize model efficency (Dai et al., 2020; Nawrot et al., 2022; 2023; Sun et al., 2023) or to skip the tokenization step of preprocessing and develop an end-to-end model (Clark et al., 2022; Tay et al., 2022; Cao, 2023; Behjati & Henderson, 2023; Behjati et al., 2023). Our research aligns with these developments by also focusing on language unit discovery. However, it differs in that these units are semantically grounded to vision.

There is also another line of work that shows language models intrinsically integrate multiple token embeddings into more meaningful semantic entities (Tenney et al., 2019; Elhage et al., 2022; Ferrando & Voita, 2024; Kamoda et al., 2025; Kaplan et al., 2025), a process referred to as detokenization. In particular, Ferrando & Voita (2024) demonstrate that specific attention heads promote the merging of subword tokens, suggesting that models actively integrate fragmented lexical units during processing. Similarly, Kaplan et al. (2025) show that language models internally augment subword tokens into coherent word-level representations, with semantic information consolidated at the final subword position. Together, these findings indicate that models naturally reorganize their inputs into semantically meaningful units beyond the granularity imposed by the tokenizer. This evidence supports the view that semantically coherent units are beneficial for representation learning though they do not promote explicitly learned, contextually grounded grouped representations which our work does.

## 4 Experiments

In this section, we empirically evaluate our proposed model. First, we probe the fine-grained vision-language understanding of our proposed text encoder under two benchmarks in Section 4.2. We show the effectiveness of our model in finding meaningful units by visualizing the attention maps in Section 4.3. We then evaluate the quality of the discovered segments quantitatively by their accordance with human-annotated groundable phrases in Section 4.4. Finally, we analyze the contributions of different aspects of our model with a series of ablation studies in Section 4.5.

### 4.1 Experimental Setup

**Datasets:** We trained our models on the training split of GCC3M dataset which consists of around 3 million image-caption pairs collected from the web (Sharma et al., 2018). We explain the datasets we used for evaluation in their corresponding sections and provide additional details in the Appendix.

**Parameters:** We first resize the images to 224×224 and then divide them into patches of size 16×16. The image encoder has 12 Transformer encoder layers with the hidden dimension of 384 and two grouping blocks at the 6th and 9th layers. The number of groups in the first block is 64 and 8 in the second block. We load the weights from the GroupViT released checkpoint[3] (Xu et al., 2022) and keep it frozen during training.

For the text encoder, we have 6 Transformer encoder layers followed by a grouping block[4] and then another 3 Transformer encoder layers. Each self-attention layer has 4 heads. Since the grouping vectors are learnable, the number of them needs to be fixed during training, so we treat the number of groups $K$ as a hyperparameter and tune it for different datasets. We experiment with $K = 1, 2, 4, 8, 16$ as the number of groups and report the performance and results of the model trained with 4 groups as it has the best performance, and study the effect of having different numbers of groups in our ablations (Table 5). The text decoder has only 1 Transformer decoder layer consisting of one self-attention and one cross attention layer, each with 1 attention

---

[3]We take the checkpoint trained on GCC3M (Sharma et al., 2018), GCC12M (Changpinyo et al., 2021) and YFCC14M (Thomee et al., 2016) datasets.

[4]Our preliminary experiments with two blocks did not lead to reasonable results.

| Model | subj | verb | object | overall |
|---|---|---|---|---|
| random | 50 | 50 | 50 | 50 |
| groupvit | 81.6 | 77.3 | 91.7 | 81.0 |
| baseline | $79.44 \pm 0.09$ | $69.52 \pm 0.35$ | $89.12 \pm 0.46$ | $75.24 \pm 0.28$ |
| ours (4 groups) | $\mathbf{80.02} \pm 0.16$ | $\mathbf{70.10} \pm 0.24$ | $\mathbf{89.92} \pm 0.45$ | $\mathbf{75.84} \pm 0.17$ |

Table 1: The zero-shot pairwise ranking accuracy of different models on the test-split of SVO probes averaged over 5 random seeds.

head. We tie the weights between the token embeddings in the encoder and the decoder. Both the encoder and the decoder have a model dimension of 128. The linear projection heads map each modality's feature vector to 256 dimension. We fix the $\tau$ to 0.07 in our contrastive losses and $\lambda$ equals to 1. We use Byte Pair Encodings (Sennrich et al., 2016) as our tokenizer with a vocabulary size of around 50k tokens and the maximum number of tokens is set to $M = 77$ following previous work (Radford et al., 2021; Xu et al., 2022). We train our models with a batch-size of 4096 for 30 epochs and use the GradeCache library (Gao et al., 2021) to obtain this batch size on a single RTX3090 GPU[5]. We trained our models with AdamW optimizer (Loshchilov & Hutter, 2019) with a learning rate of 0.0016 with linear warmup for 2 epochs and cosine annealing decay.

**Baselines:** In order to showcase the effect of discovering groups as part of the architecture in vision and language models, we consider the following baseline. We replace our proposed Text Group Transformer with a vanilla Transformer encoder that has 9 Transformer encoder layers, thereby having approximately the same number of parameters as our model (referred to as *baseline* in the experiments). This text encoder architecture is the one used in GroupViT and other dual-stream vision-language models like CLIP (Radford et al., 2021). We train it under the same training setup as our own model to provide a fair comparison. That is, we freeze the image-encoder and only train the text side from scratch with the same training data and objectives as our models. We take the final text representation from the encoded <eos> token.

In addition, we report the results of the trained GroupViT model with its own text encoder (i.e., 12 Transformer encoder layers) and 2 layer projection heads (referred to as *groupvit* in the experiments). Note that this model has many more parameters and has been trained on 10x more data.

### 4.2 Fine-grained Vision-Language Understanding Probes

We evaluate the fine-grained vision and language understanding of our model by employing different benchmarks which are specifically designed for this purpose. We will explain each of these benchmarks and the zero-shot performance of our models in the following sections. In each case, the zero-shot classifier ranks the image-text pairs by their similarity scores $\text{sim}(\hat{z}_j^T, \hat{z}_i^I)$, which is the cosine between the pooled embeddings on the image and text sides. We refer to the score obtained from this zero-shot classifier as pair-wise ranking accuracy.

#### 4.2.1 SVO Probes

Hendricks & Nematzadeh (2021) designed a benchmark where they pair every sentence with two images, one positive and one negative. The negative images are selected in a controlled fashion where only either subject, verb or the object of the image is different from the original one.

Table 1 shows the results of the zero-shot performance of different models under this benchmark. We observe that our model significantly outperforms the transformer baseline model in overall accuracy, achieving a mean improvement of 0.60 percentage points across five random seeds (paired t-test, $p = 0.013$). Improvements are significant for subject and object accuracy, while verb accuracy shows a positive but non-significant trend. The results verify our hypothesis that representing the language in a fine-grained and meaningful

---

[5]It takes around 48 GPU hours for every model to train.

| Model | accuracy |
|---|---|
| random | 50 |
| groupvit | 82.5 |
| baseline | $80.56 \pm 0.51$ |
| ours (4 groups) | $\mathbf{81.25} \pm 0.44$ |

Table 2: The zero-shot performance of different models on the test-split of FOIL-COCO benchmark averaged over 5 random seeds.

| | SVO | | | | |
|---|---|---|---|---|---|
| Model | subject | verb | object | overall | FOIL-COCO |
| without finetuning | 80.2 | 70.10 | 89.92 | 75.84 | 81.25 |
| finetuning the last layer of the vision encoder | **81.4** | **70.3** | 89.9 | **76.2** | **82.69** |
| finetuning the whole vision encoder | **81.4** | **70.3** | **90.0** | **76.2** | 82.66 |

Table 3: The performance of different models with/without fine-tuning the vision encoder for the last 4 epochs of training.

manner helps the fine-grained vision and language understanding of the model. Both of these models are well above the random baseline. The results for GroupViT's Transformer model are not comparable because it is trained on much more data, but we see that the resulting increase is much higher on verbs than on the the groundable phrases (subjects and objects) that our model is designed to represent as separate vectors.

### 4.2.2 FOIL-COCO

Shekhar et al. (2017) propose FOIL-COCO dataset where for every image there is a correct caption and a "foil" one. The foil caption is different from the original caption by altering one of the nouns in the original caption into a foil one. In Table 2, we observe that our model demonstrates a good performance, outperforming the transformer model in mean performance (paired t-test, $p = 0.17$). This indicates that the noun understanding of our model has improved by learning fine-grained representations. Additionally, despite being trained on substantially less data than the GroupViT text encoder, our model performs nearly as well.

### 4.2.3 Finetuning the Vision Encoder

In order to demonstrate the potential upper bound of our model's fine-grained vision-language understanding, we unfreeze the vision encoder in the last epochs of training. Specifically, we take our model's checkpoint at epoch 26 and further finetune the vision and language sides with a learning rate of 4e-5 for 4 more epochs. We try two setups: first, finetuning the last layer of the vision encoder and second, finetuning the whole model.

Table 3 shows the results on the SVO probes FOIL-COCO dataset. We observe that the performance of our model has improved on the vision and language understanding tasks by finetuning the vision side in both experiments. This allows the model to reduce the contrastive loss by tuning the parameters on the vision side while keeping the reconstruction loss low. Therefore, the model has the potential to learn better alignments between vision and text when trained on both sides.

### 4.3 Attention Visualization

In order to understand what each group is representing, we visualize the soft attention weights of the groups over the input tokens in Figure 2. Interestingly, we observe that contiguous segments have emerged, without imposing any contiguity constraints in the groupings. We believe that this is due to the fact that usually in language the contiguous tokens capture highly correlated information and that's why our model is grouping

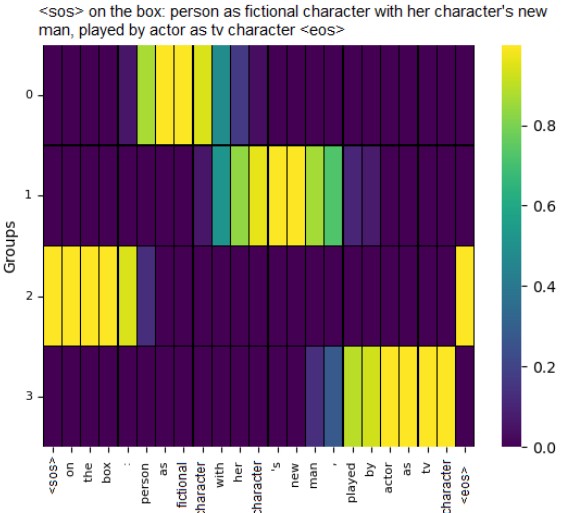

Figure 2: Soft attention of the groups over the input tokens. It shows that contiguous segments have emerged which capture phrase-like units.

them together as part of its compression. Moreover, we can see that the emerging segments are meaningful in that they capture phrase-like units. We quantitatively evaluate the phrase discovery performance of our model in Section 4.4. In our examination of a sample of attention maps, we observe that a given group tends to bind to similar positions in the text, but that the boundaries between groups vary.

### 4.4 Zero-shot Segmentation Evaluation

In order to evaluate the emerging segments in the attention maps quantitatively, we compare to human-annotated groundable phrases using several metrics (Table 4). For the gold segmentation, we use the annotations in Flickr30k Entities (Plummer et al., 2015) where groundable phrases are human-annotated. We report the results on the validation set of this dataset.

We propose a metric similar to Intersection-over-Union (IoU) in the visual object detection literature which we call "tIoU". We first compute the soft attention weights of the groups over the input tokens. Then, by taking the argmax over the inputs, we have an assignment matrix of every input to a group. Given a gold segmentation, we can compute the IoU for each discovered group of tokens and each gold segment. For the computation of IoU, the intersection is equal to the number of overlapping tokens. For the union, we do not count the tokens which were not annotated in the dataset, as the annotators did not have the constraint to include all the tokens in their annotation. This gives us a matrix where by applying the Hungarian matching algorithm (Kuhn, 1955) maximizing this metric, we can obtain a 1-1 mapping between the discovered groupings and the gold segments. By having the mappings, we can compute precision, recall and F1 as well as IoU for each paired group and gold segment. In reporting the results, we first average every metric for the text input and then, report the average over all examples.

In Table 4, we report the results of our evaluation. We compare our model against multiple baselines, including an untrained, randomly initialized text group transformer model. We also report the performance of applying different clustering methods over the encoded features of our transformer baseline. In particular, we apply k-means, spectral clustering (Shi & Malik, 2000) and mean shift (Comaniciu & Meer, 2002) with 4 clusters. We observe that our model surpasses all the baselines by a large margin in almost all the metrics. Specifically, the high tIoU indicates that our model is indeed very good at discovering groundable phrases in the captions.

| Model | tIoU | P | R | F1 |
|---|---|---|---|---|
| random | 42.15 | 61.51 | 60.03 | 54.54 |
| baseline (k-means) | 52.77 | 61.82 | 64.87 | 59.55 |
| baseline (spectral-clustering) | 38.88 | 49.81 | 52.82 | 45.52 |
| baseline (mean shift) | 50.38 | **99.64** | 51.73 | 65.13 |
| ours (4 groups) | **76.42** | 87.25 | **85.83** | **83.72** |

Table 4: Phrase segmentation performance of different models under different evaluation metrics.

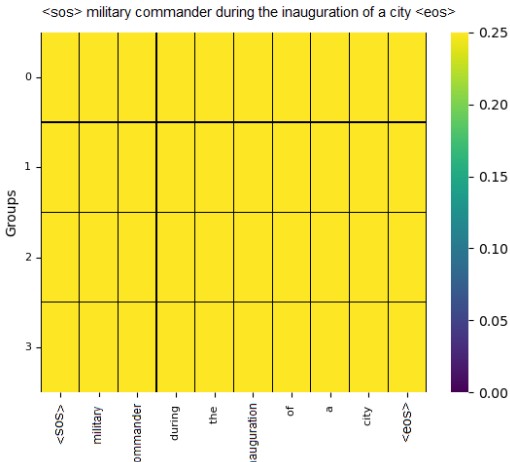

Figure 3: Soft attention of the groups over the input tokens for a model trained without the reconstruction loss. It shows a uniform attention map and lack of segmentation.

### 4.5 Ablation Study

In this section, we study the effect of different design choices (i.e., training losses and number of groups) on the performance of our models both in terms of groundable phrase discovery and fine-grained vision and language understanding.

### 4.5.1 Training Losses

In Table 5 we see the different effects of the two types of loss on our model. Without the contrastive loss, the model has no training on the image-text relationship, so it is not surprising that the image-text semantic evaluations are very low. More surprisingly, although it still segments in a meaningful way without contrastive loss, the segmentation corresponds slightly less well to groundable phrases[6]. This suggests that semantic grounding in images actually helps the model discover meaningful units of text.

Interestingly, without the reconstruction loss, the model fails to segment in a meaningful way. We can see this both in the tIoU score and in the uniform attention pattern shown in Figure 3. This lack of segmentation in turn affects the fine-grained understanding of the image-text relationship. The holistic representations indicated by Figure 3 are relatively good at representing verbs, because verb understanding combines information across multiple objects. But if we only consider the noun phrases (i.e. subjects, objects categories from SVO probes and FOIL-COCO), averaged in the last column, then segmenting the representation according to semantic objects, as indicated in Figure 2, results in significantly higher understanding of the image-text relationship (paired t-test, $p < 0.001$). Additionally, we demonstrate that the reliance on recon-

---

[6]The difference is not statistically significant (paired t-test, $p = 0.14$)

| Model | SVO | | | | FOIL-COCO | Noun Understanding |
|---|---|---|---|---|---|---|
| | subject | verb | object | overall | | |
| ours | $\mathbf{80.02} \pm 0.16$ | $70.10 \pm 0.24$ | $\mathbf{89.92} \pm 0.45$ | $75.84 \pm 0.17$ | $\mathbf{81.25} \pm 0.44$ | $\mathbf{83.74} \pm 0.30$ |
| w/o contr loss | $50.60 \pm 0.39$ | $50.06 \pm 0.55$ | $50.48 \pm 0.94$ | $50.26 \pm 0.35$ | $49.99 \pm 1.48$ | $50.47 \pm 0.75$ |
| w/o rec loss | $78.74 \pm 0.63$ | $\mathbf{72.36} \pm 0.23$ | $88.88 \pm 0.15$ | $\mathbf{76.84} \pm 0.15$ | $77.69 \pm 0.26$ | $81.78 \pm 0.12$ |

| Model | tIoU |
|---|---|
| ours | $\mathbf{76.42} \pm 2.28$ |
| w/o contr loss | $73.18 \pm 1.83$ |
| w/o rec loss | $40.52 \pm 2.01$ |

Table 5: The performance of our model compared to the ablated ones on multiple datasets averaged over 5 random seeds. Noun understanding refers to the average of performance on noun phrases (i.e. subjects, objects and FOIL-COCO).

| # of groups | tIoU | SVO | FOIL-COCO |
|---|---|---|---|
| 1 | 43.55 | 74.99 | 80.23 |
| 2 | 53.12 | 75.30 | 80.01 |
| 4 | $\mathbf{76.42}$ | $\mathbf{75.84}$ | $\mathbf{81.25}$ |
| 8 | 63.93 | 74.8 | 80.56 |
| 16 | 52.54 | 72.4 | 79.43 |

Table 6: The performance of our model trained with different number of groups.

struction loss is not a byproduct of the randomly initialized text encoder in Appendix B. Therefore, both losses are crucial for the purpose of discovering groups of tokens with groundable semantics.

### 4.5.2 Number of Groups

In Table 6, we investigate the effect of training our model with different numbers of groups. We can see that the model trained with 4 groups achieves the best results in all our evaluations. This implies that having too many or too few groups hurts the performance of our model and this hyperparameter should be tuned for different datasets and tasks.

## 5 Conclusions

In this work, we developed a novel model for discovering meaningful units that are semantically aligned to the objects in the image. We freeze an image encoder which outputs groups that approximately represent objects and employ an analogous architecture on the text side to discover units that are at the level of phrases. While many dual-stream VLMs represent text as a single vector, we hypothesize that learning to represent language at a finer granularity will improve their fine-grained vision and language understanding.

We verified our hypothesis by employing two specifically designed probing benchmarks, namely, SVO probes and FOIL COCO. In addition, we showed that the segments that appear in the attention maps of groups attending to tokens are meaningful both qualitatively and quantitatively, in terms of overlap with human-annotated groundable phrases. Moreover, we ablated the effect of our losses on learning these units and concluded that both are necessary for having meaningful and semantically aligned units.

Our experiments reveal the potential of our proposed model in specific real-world downstream applications such as semantic segmentation and concept binding, where fine-grained knowledge is essential.

## Limitations

We have performed our experiments on the datasets and benchmarks in English. However, we do not make any language dependent assumptions in developing our model. Therefore, we believe that our method is generalizable across other languages as long as enough data for training is available.

We were not able to perform our experiments at scale due to the computational limitations. We expect that training the image and text encoder simultaneously from scratch would lead to better alignment between the two modalities, which should in turn improve our results.

Lastly, our model by design is limited to a fixed number of groups, which is a common limitation in many object discovery works, such as Xu et al. (2022); Singh et al. (2022); Seitzer et al. (2023). Therefore, for every dataset and task, it should be tuned as a hyperparameter for optimal performance. While this brings additional costs, it reduces the complexity of the problem we are tackling. Namely, the multi-modal setting is already very complicated and we need to make this simplifying assumption in order to make progress on the essential problem of finding groupings at all. Dynamically choosing the number of groups is a topic for future research.

## Acknowledgments

This project was partly supported by the Swiss National Centre of Competence in Research (NCCR) under the project Evolving Language, grant number "51NF40_180888".

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

## A    Details of the Datasets

We provide additional details about the datasets we used in our work in the following.

- GCC3M: The average caption length in this dataset is 10.5 tokens.

- SVO probes: The test split of this dataset contains around 30k examples.

- FOIL-COCO: The test split of this benchmark which has around 99k examples.

- Flickr30k Entities: The validation set of this dataset has around 5000 examples. The number of annotated phrases in this dataset is on average 3.5.

## B    Additional Experiments with Pretrained Text Encoder

We conducted two experiments to verify if the reliance on reconstruction loss is a byproduct of the randomly initialized text encoder. We first trained a text encoder (a Transformer Decoder with 9 layers and model dimension of 128) on Wiki103 dataset (Merity et al., 2017) with the next token prediction objective. We then loaded the weights of its first 6 layers to our Text Group Transformer; i.e., all the layers before the grouping block. Afterwards, we train our model with only the contrastive loss in the same setup as before. In addition to this experiment, we also tried loading the weights of the first 6 layers of the text encoder of GroupViT (Xu et al., 2022) and trained it with only the contrastive loss. Both of the trained models showed the same behavior as starting from random. Particularly, the groups' attention map showed a nearly uniform distribution over the tokens. We perceive that learning a single-vector representation is generally easier than learning a meaningful distributed representation, and the model tends to learn that. Additionally, since we are using the frozen vision encoder, the representations we get from the vision side were previously aligned with a single vector representation of text. Therefore, recovering that single-vector representation is an easier task for the model to learn in order to decrease the loss (which is indeed the same as what the model has been trained on), without the need for distributing the tokens to the groups. Reconstruction loss encourages the groups to capture distinct segments of data so that it can better decode the sequence and thus, it is a necessary part of our training objectives.

## C    Artifacts statements

The datasets used do not have personally identifying information or offensive content. We provide the list of datasets used and the corresponding licenses in Table 8, which are all consistent with our academic use.

## D    Descriptive Statistics

We report the results in Table 1, Table 2, and Table 5 after training the models with five random seeds. Other results are from single runs.

## E    Packages

We provide a list of packages used in our code in Table 7.

## F    AI Assistants

We utilized AI assistants for minor text editing and code completion tasks during the development of the model.

| Package | version |
|---|---|
| Python | 3.7 |
| PyTorch | 1.8 |
| webdataset | 0.1.103 |
| mmsegmentation | 0.18.0 |
| timm | 0.4.12 |
| nltk | 3.8.1 |
| ftfy | 6.1.1 |
| regex | 2023.6.3 |

Table 7: The packages used in our code development
.

| Dataset | License |
|---|---|
| GCC3M | Google license (link) |
| SVO-Probes | Creative Commons Attribution 4.0 International Public License (CC BY 4.0) |
| FOIL-COCO | Creative Commons Attribution 4.0 License |
| Flikr | Creative Commons Attribution 0: Public Domain |

Table 8: Datasets and their licenses.

