# OpenReview forum: "Discovering Meaningful Units with Visually Grounded Semantics from Image Captions"
_TMLR — Accepted by TMLR_

### Review · Reviewer_Zr52 · 2026-01-19

**Summary Of Contributions:**

The paper proposes to learn a set based representation for text captions accompanying images. The goal of the set based representation is to group phrases referring to objects into a single representations. To achieve this the paper introduces a text-group transformer which learns a learnable clustering of the text tokens following the ideas from slot attention. Furthermore, the approach utilizes a contrastive and a reconstruction loss to 1) align the the text representations to the image features and 2) learn meaningful text groupings. The resulting model is applied to two vision language understanding tasks - SVO probes and FOIL COCO - showing stronger performance to a simple transformer based text encoder where the set based representation is replaced by a single global representation for the whole text.

strengths:

1) The idea of the paper seems interesting. To my knowledge, this is the first paper to explore the idea of grouped representations for language which has been common for vision in the past
2) The method is simple and easy to understand and from my understanding, does not require a lot of hyperparameter tuning which has been one of the pain points for some visual object centric models

weaknesses
1) I am not fully convinced of the utility of such grouped representations for text as opposed to simple token-level representations (I will detail my concerns below)
2) I think the baselines are not extensive. Currently the only baseline in the vision language understanding experiments is the parameter-matched transformer. I think to further understand the effectiveness of the proposed approach the authors should consider adding more thorough baselines (detailed below)

Minor comment:
In section 2, the word "subword" is repeatedly used to denote tokens in language. Maybe you can simply use the word "tokens" since it is well established terminology in language and vision-language communities.

**Additional Comments:**

Questions -

1) The limitation section highlights that having a fixed number of groups can be be detrimental especially as we move to complex captions with varying number of objects. This has been a valid limitation for many years in the object-centric community and even now there is not clear solution. Given this, do the authors think it is worth investing more time in such ideas or would simple token-level approaches win as caption complexity increases.

**Audience:**

Yes

**Audience Explanation:**

I think the paper would be of interest to people in the TMLR audience as to my knowledge it is the first paper to explore structured representations for text. The findings of the paper, whether in favour of such representations or not, will be useful to the community and would provide valuable insights for people working in this area.

**Broader Impact Concerns:**

No ethical concerns

**Claims And Evidence:**

No

**Claims Explanation:**

The claim that the paper makes is that the resulting set-based representation of language and allows the model to have a fine-grained understanding of vision-language which in-turn should help vision language understanding tasks. I am not fully convinced that the experiments support this claim as looking at the results in Table 1 and 2, the advantage of the proposed approach is very small compared to the transformer-based baseline and the results and no standard deviation number are reported. Moreover, looking at table 4, the runs without the reconstruction loss are not visibly worse than the ones with both reconstruction loss + contrastive loss (again only 1 run is reported hence it is hard to tell).

Building up on the previous point, I think there should be more baselines in comparison to conclude that such structured representations for language help. For instance, the third baseline considered is the groupvit language encoder but the authors say it is not a fair comparison since it is trained on a lot more data. I wonder if its possible to train groupvit like langauge encoder at their scale of data for a fair comparison. Group ViT also has a similar contrastive loss as used in this paper and additionally it also has a multi-label image text contrastive loss. Both these losses, I believe can be run at the scale available to the authors.

I believe both my concerns point to the main issue that I don't find enough evidence in the paper to show that discovering such representations for language help in vision-language understanding tasks.

**Requested Changes:**

1) Firstly, the authors should report each results over multiple seeds to see if the improvements are statistically significant. I think this is critical for acceptance.
2) Depending upon whether, the improvements of the proposed technique are statistically significant, the authors should temper the claims of the paper i.e. if the improvements are not statistically significant, the authors should not claim that these representations are useful for vision-language understanding tasks.
3) the authors should consider adding more baselines such as the one highlighted above about group-vit.

---

> ### Author Response · Authors · 2026-02-11
> **Response to Reviewer Zr52**
>
> > 1,2. The authors should report each results over multiple seeds to see if the improvements are statistically significant
>
> As the reviewer requested, we run our experiments with 5 seeds, and here are the updated tables for the SVO probes and Foil-COCO probes. We updated the draft accordingly.
>
> #### Accuracy (%) averaged over 5 random seeds for SVO probes.
> | Model | Subj | Verb | Obj | Overall |
> |:------|-----:|-----:|----:|--------:|
> | ours (4 groups )  | 80.02 ± 0.16 | 70.10 ± 0.24 | 89.92 ± 0.45 | 75.84 ± 0.17 |
> | baseline  | 79.44 ± 0.09 | 69.52 ± 0.35 | 89.12 ± 0.46 | 75.24 ± 0.28 |
>
> For SVO probes, our model significantly outperforms the baseline model in overall accuracy, achieving a mean improvement of 0.60 percentage points across five random seeds (paired t-test, p = 0.013).
>
> #### Accuracy (%) averaged over 5 random seeds for Foil-COCO probes.
>
> | Model         | Accuracy (mean ± std) |
> |---------------|-----------------------|
> | foil_our_4g   | 81.25 ± 0.44 |
> | foil_base     | 80.56 ± 0.51 |
>
> For Foil-Coco, our model achieves better mean performance (+0.69 percentage) than the baseline (paired t-test, p = 0.17).
>
> Overall, the results are consistent with the claims we made, showing our proposed architecture achieves better vision-language understanding over a CLIP-style text encoder (i.e., single-vector representation).
>
> > Moreover, looking at table 4, the runs without the reconstruction loss are not visibly worse than the ones with both reconstruction loss + contrastive loss
>
> Due to limitations in compute, we don’t have the results ready for multiple seeds for Table 4. We will include them in the camera-ready version of the paper. We would like to note that while improving the vision language understanding is one of the goals of our paper, discovering meaningful units as part of the architecture is also an important aspect of our work. The model trained without the reconstruction loss does not show any meaningful patterns learnt in the attention maps of groups over the tokens (uniform distribution for the attention of groups over the tokens).
>
> > 3. the authors should consider adding more baselines such as the one highlighted above about group-vit.
>
> Firstly, just for clarification, the baseline which we report is a version of the GroupViT which is directly comparable to our model in our experimental setup.  But the reviewer is correct that this baseline does not include the multi-label image-text contrastive loss.  The multiple labels used in this loss are generated with templates which rely on hand-coded definitions of groundable phrases.  Because our objective is to learn groundable phrases automatically, we do not think that adding this loss would address whether our method succeeds in finding useful groundable phrases.
>
> > Minor comment: In section 2, the word "subword" is repeatedly used to denote tokens in language.
>
> Thank you for your suggestion. We updated the draft accordingly.
>
> > Answer to the Question 1.
>
> We believe yes, it is still an interesting research question and worth investing time in it. In addition to the scientific contribution,  this line of work brings interpretability value to the community as we develop models that are more interpretable by design. There has also been some work on trying to dynamically decide the number of groups by encouraging the use of fewer groups with sparsity-imposing losses, as in [1,2,3], which could be combined with the methods developed in this paper.
>
> [1] Gopalakrishnan, A., Irie, K., Schmidhuber, J., & van Steenkiste, S. (2023). Unsupervised learning of temporal abstractions with slot-based transformers. Neural Computation.
>
> [2] Behjati, M., & Henderson, J. (2023). Inducing meaningful units from character sequences with dynamic capacity slot attention. TMLR.
>
> [3] Behjati, M., Fehr, F., & Henderson, J. (2023). Learning to abstract with nonparametric variational information bottleneck. Findings of EMNLP.

---

### Review · Reviewer_USZ4 · 2026-01-28

**Summary Of Contributions:**

The authors suggest a way to align image patch groups with caption token groups at a high (object-specific) level, by grouping textual caption tokens to groups. They measure the performance of this method on two downstream tasks, and perform some analysis of attention patterns to visualize the internals of their method.

**Audience:**

Yes

**Audience Explanation:**

Yes, as this paper researches the question of vision-to-text alignment in contrastive VLMs from a (very) specific perspective.

**Broader Impact Concerns:**

None.

**Claims And Evidence:**

No

**Claims Explanation:**

First, my main problem is with the first contribution listed in the paper: the authors state they "develop a novel model" but all of its components are based on an existing method (Xu et al. in the paper). Thus it is unclear what is the main contribution made by the authors here, making the analysis of its validity irrelevant at the current state.

Second, the authors state they compare their method to single-vector text-encoding models (such as CLIP), and reach a better fine-grained vision and language understanding. However, the results show otherwise, showing better results for the "groupvit" baseline. The authors state that "The results for GroupViT’s Transformer model are not comparable because it is trained on much more data" - but this is not relevant in my opinion. One can't limit the baselines comparison to a very narrow setting that only includes the presented method.
Additionally, the usage of only two specific benchmarks, while a plethora of other benchmarks for object relations in VLMs exist, makes the results seem less reliable.

**Requested Changes:**

1. I would like the contributions to be more clearly - it seems the learnable group vectors are adapted from previous work (Xu et al.; which also contribute the architecture of the image encoder and other components). Thus it is unclear what is novel in the presented model (namely in the first contribution "We develop a novel model to discover meaningful units…").

2. To make the results more beliveable, I would like to see further downstream tasks and other baselines. Also, I would suggest against merely ignoring the better results of the groupvit baseline on the claim that "its not a fair comparison" and instead find the benchmarks or cases in which the suggested method is better.

3. There is a long line of studies on detokenization (e.g., [1] and it's related work section) and general lexicon learning in LLMs - analyzing how LLMs implicitly group separate tokens to meaningful "objects". I think a connection to this line of work will help, by explaining which additional value the proposed method shows that doesn't implicitly exist in LLMs that do detokenization. Maybe this can come in the "unit discovery" section of the related work.

4. Generally, the writing is over-verbose and repetitive at times, and it would make the paper more clear if it was written more concisely.

5. The writing is also missing technical details. For example, 2.1.1 states "pass the resulting vectors through some transformer layers" - how many is some? Please address this and similar points throughout the paper.

---

> ### Author Response · Authors · 2026-02-06
> **Response to Reviewer USZ4**
>
> We thank the reviewer for their feedback.
>
> > 1. It is unclear what is novel in the presented model
>
> It is true that we have borrowed our model’s architectural components from the previous work (i.e., Xu et al.), but we have adapted the model to a new domain (i.e., language) and defined a new task (i.e., phrase discovery in language). In particular, we proposed different training objectives from Xu et al. to specifically address the challenges of dealing with language. Therefore, we believe our approach is novel both relative to the model of Xu et al. and relative to other work in the domain of language, as Reviewer Fdbq and Zr52 have also mentioned.
>
> > 2. I would like to see further downstream tasks and other baselines.
>
> Our “baseline” model is exactly the GroupVIT model, but trained in a comparable way to our model.  We insist that comparing two models trained on vastly different amounts of data is not a valid scientific comparison.  We do not claim that training on less data is better; we claim that our architectural changes are better.  So we provide a scientific comparison with and without our architectural changes, with all other things being equal.
> So that we can respond to this point in more detail, we would appreciate it if the reviewer could provide some specific suggestions for the additional downstream tasks and baselines.
>
> > 3. There is a long line of studies on detokenization (e.g., [1] and its related work section).
>
> We thank the reviewer for pointing this out. We will include this line of work in our related work section. What is [1] referring to? Is it “From Tokens to Words: On the Inner Lexicon of LLMs”?
>
> > 4. The writing is over-verbose and repetitive at times.
>
> We would appreciate it if the reviewer could be more specific and provide examples of where the repetitions and over-verbosity have occurred, so that we can fix them.
>
> > 5. The writing is also missing technical details. For example, 2.1.1 states "pass the resulting vectors through some transformer layers" - how many is some?
>
> We have explained the general architecture design in Section 2, and have our implementation details in Section 4.1 (parameters). As for the example the reviewer has mentioned, the number of layers is stated in Section 4.1 (Parameters), i.e., 6 transformer encoder layers. What other details does the reviewer think are missing?

---

> > ### Author Response · Authors · 2026-02-11
> > **Response to Reviewer USZ4**
> >
> > > 2. I would like to see further downstream tasks and other baselines.
> >
> > We have conducted further experiments according to other reviewers' suggestions, notably using pre-trained weights for the models, fine-tuning the vision side of the model, and re-running the experiments with multiple seeds. Please refer to the responses to other reviewers for the discussion.
> >
> > > >3. There is a long line of studies on detokenization (e.g., [1] and its related work section).
> >
> > >We thank the reviewer for pointing this out. We will include this line of work in our related work section. What is [1] referring to? Is it “From Tokens to Words: On the Inner Lexicon of LLMs”?
> >
> > We have added the following discussion about detokenization to the related work section of the paper.
> >
> > There is also another line of work that shows language models intrinsically integrate multiple token embeddings into more meaningful semantic entities [Tenney et al 2019, Elhage et al 2022, Ferrando et al 2024, Kamoda et al 2025, Kaplan et al 2025], a process referred to as detokenization . In particular, [Ferrando et al 2024] demonstrate that specific attention heads promote the merging of subword tokens, suggesting that models actively integrate fragmented lexical units during processing. Similarly, [Kaplan et al 2025] show that language models internally augment subword tokens into coherent word-level representations, with semantic information consolidated at the final subword position. Together, these findings indicate that models naturally reorganize their inputs into semantically meaningful units beyond the granularity imposed by the tokenizer. This evidence supports the view that semantically coherent units are beneficial for representation learning though they do not promote explicitly learned, contextually grounded grouped representations which our work does.

---

### Review · Reviewer_Fdbq · 2026-01-29

**Summary Of Contributions:**

The paper proposes a novel approach to enhance the fine-grained understanding of dual-stream Vision-Language Models (VLMs), specifically CLIP-style architectures, by discovering meaningful semantic units through token grouping on the text side. The key contributions are summarized as follows:

1. High Research Value and Novel Perspective. While most of the existing works on enhancing fine-grained VLM capabilities focuses on re-designing or training the vision side while freezing the text encoder , this work argues that the text side might actually be the bottleneck. By shifting the focus to discovering groundable phrases in language, the paper addresses a crucial yet under-explored area.

2. Architectural Adaptation. This work adapts the grouping mechanism, which originally designed for vision transformer from GroupViT, to a language Transformer. This allows the model to learn representations at the level of objects rather than individual subwords or single-vector global embeddings at the text side.

3. Refined Training Objectives. Beyond the standard contrastive loss, the paper introduces a simple text reconstruction loss. This objective encourages the learned groups to retain sufficient information to reconstruct the original caption, ensuring that the discovered units are both compact and semantically rich.

**Additional Comments:**

My overall assessment is relatively favorable. I leave it to the Editor to consider my feedback and arrive at a final decision. Thank you.

**Audience:**

Yes

**Audience Explanation:**

While the technical design of this paper does not introduce substantial architectural innovation, it represents a meaningful research topic by applying fine-grained enhancement strategies, which are predominantly explored on the vision side, to the text encoder. Given that the majority of current research on improving CLIP's fine-grained capabilities is concentrated on the visual side, this research motivation is inherently valuable and is likely to attract interest from some individuals in TMLR's audience focused on vision-language representation learning.

**Broader Impact Concerns:**

No concerns.

**Claims And Evidence:**

Yes

**Claims Explanation:**

The authors provide sufficient evidence to support their primary claims through both quantitative benchmarks and qualitative analysis:

1. Effectiveness of Group Representations.

To verify if grouping tokens outperforms the traditional single-vector <eos> representation , the authors evaluate the model on the SVO Probes and FOIL-COCO benchmarks. The results in Table 1 and Table 2 demonstrate that the proposed model achieves better overall performance in fine-grained tasks, particularly in noun and object understanding, compared to the vanilla Transformer baseline.

2. Discovery of Meaningful Units.

To address whether the groups learn groundable semantic units, the paper presents attention maps in Figure 2. These visualizations show that the model naturally discovers contiguous segments corresponding to phrase-like units without explicit contiguity constraints. This is further supported by quantitative tIoU metrics on the Flickr30k Entities dataset, where the model significantly outperforms clustering-based baselines.

**Requested Changes:**

Overall, I find the experimental evaluation in this submission to be relatively weak. To improve the technical soundness and depth of the paper, I request the authors address the following concerns:

1. The ablation study shows that without the reconstruction loss, the group tokens lose their ability to capture meaningful textual units, resulting in a uniform attention map. This result is somewhat counter-intuitive. I suspect this failure may be a consequence of training the text encoder from a random initialization. Please clarify if the reliance on reconstruction loss is a byproduct of the randomly initialized encoder. I recommend conducting an experiment using a pre-trained text encoder to see if it can discover meaningful units via grounding alone, without the reconstruction loss.

2. The current ablation study indicates that the model performs best when the number of groups is set to K=4. However, there is a lack of analysis regarding the relationship between K and the complexity or length of the input text. Theoretically, longer and more descriptive captions should require a higher number of group tokens to capture all distinct semantic units. Therefore, it is necessary to use traditional linguistic metrics to measure the complexity of the training text and verify if increasing K leads to better performance on more complex subsets of the data.

3. The current methodology keeps the vision encoder frozen throughout the entire training process. While this reduces computational cost, it likely limits the ultimate alignment between modalities. I suggest the authors to provide experimental results where the vision encoder is unfrozen during the final stages of training. This would demonstrate the potential upper bound of the model's fine-grained alignment capabilities.

4. Lack of statistical results regarding the attentions of group tokens. Figures 2 and 3 present specific qualitative examples of attention maps to argue that group tokens learn effective textual units. While these are illustrative, they do not provide a complete picture of the model's behavior across the entire dataset and could be interpreted as "cherry-picked" examples. It is better to please provide more comprehensive quantitative or statistical evidence to demonstrate that these meaningful groupings are a consistent property of the model across a diverse range of samples, rather than isolated cases.

---

> ### Author Response · Authors · 2026-02-11
> **Response to Reviewer Fdbq**
>
> We thank the reviewer for taking the time to provide feedback on our work. We are glad that the reviewer finds high research value in our paper and is generally positive about our work. We address the reviewer requests in the following.
>
> > 1. I recommend conducting an experiment using a pre-trained text encoder to see if it can discover meaningful units via grounding alone, without the reconstruction loss.
>
> We thank the reviewer for their suggestion. We conducted two experiments to explore this direction. We first trained a text encoder (a Transformer Decoder with 9 layers and model dimension of 128) on Wiki103 dataset with the next token prediction objective. We then loaded the weights of its first 6 layers to our GroupViT text encoder; i.e., all the layers before the grouping block. Afterwards, we train our model with only the contrastive loss in the same setup as before. In addition to this experiment, we also tried loading the weights of the first 6 layers of the text encoder of GroupViT (Xu et al.) and trained it with only the contrastive loss.
> Both of the trained models showed the same behavior as starting from random. Particularly, the groups’ attention map showed nearly uniform distribution over the tokens. We explain our intuition for this behavior in the following. Learning a single-vector representation is generally easier than learning a meaningful distributed representation, and the model tends to learn that. Additionally, since we are using the frozen vision encoder, the representations we get from the vision side were previously aligned with a single vector representation of text. Therefore, recovering that single-vector representation is an easier task for the model to learn in order to decrease the loss (which is indeed the same as what the model has been trained on), without the need for distributing the tokens to the groups. Reconstruction loss encourages the groups to capture distinct segments of data so that it can better decode the sequence and thus, it is a necessary part of our training objectives.
>
> > 2. Theoretically, longer and more descriptive captions should require a higher number of group tokens to capture all distinct semantic units.
>
> Yes, this is true. Different captions have different text lengths and complexities, and the fixed number of groups (K) limits the model’s capabilities to correctly capture the units. Having too many groups causes over segmentation, and having too few causes under segmentation. Therefore, K should be tuned as a hyperparameter for every dataset and more ideally, dynamically tuned for every sample in the dataset. The authors in [Zimmermann et al 2023] experimentally verified this point, and showed that slot-based object-centric models (like our model) are sensitive to the number of slots and it affects their downstream performance.
>
> *Zimmermann et al., Sensitivity of Slot-Based Object-Centric Models to their Number of Slots, arXiv 2023*
>
>
> >3. I suggest the authors to provide experimental results where the vision encoder is unfrozen during the final stages of training.
>
> We thank the reviewer for their suggestion. This is indeed an interesting experiment.
> We took our model’s checkpoint and further finetune the vision and language sides with a learning rate of 4e-5 for 4 more epochs. We both tried finetuning the last layer of the vision encoder as well as the whole model. We hereby report the results in the following table.
>
>
> | Setup| Foil (%) | SVO Overall (%) |
> |:-------|---------:|----------------:|
> | without finetuning | 81.25 | 75.84|
> |  finetuning the last layer of the vision encoder | 82.69 | 76.2 |
> |  finetuning whole vision encoder     | 82.66 | 76.2 |
>
>
> We can see that the performance of our model has improved on the vision language understanding tasks by finetuning the vision side in both experiments. This allows the model to reduce the contrastive loss by tuning the parameters on the vision side while keeping the reconstruction loss low. Therefore, the model has the potential to learn better alignments between vision and text when trained on both sides. We will add this experiment to the paper.
>
> > 4.  Lack of statistical results regarding the attentions of group tokens.
>
> We have already devoted Section 4.4 to the quantitative analysis of the resulting attention maps using several metrics  and provided the results in Table 3. Table 3 shows that the groupings are consistent in terms of accordance with groundable phrases, and our conclusions are not just made by the attention visualizations.

---

### Decision · Action_Editor_sCeH · 2026-03-27

**Recommendation:** Accept with minor revision

**Additional Comments:**

**Larger changes**

**Statistical significance reporting for ablations in table 4**. In this table, some of the improvements are very small (e.g. tIoU with (76.42) and without (76.18) contrastive loss), and significance should be calculated for this. One method that does not involve rerunning experiments is to use bootstrapping (https://en.wikipedia.org/wiki/Bootstrapping_(statistics)). This allows you to calculate significance quickly and easily.

**A reconsideration of the strength of the claims**. In terms of improvement of understanding, although the overall improvement on SVO Probes is significant (Table 1 and section 4.2.1), the improvements are really quite small: 0.6 percentage points. Statements like "but our multi-vector representation does a *much better* job of representing objects" are too strong. Similarly, for FOIL-COCO (Table 2, Sec 4.2.2) the phrase " our model demonstrates a *remarkably good* performance, outperforming the transformer model in mean performance (paired t-test, p = 0.17)." is too strong, given an improvement of 0.69 percentage points. I also note that p=0.17 is high, and would not usually be considered a significant improvement.

**Smaller changes**
Figure 1 has typos (endengared cheeta). If this is the actual text from the dataset, mention in the figure caption.

Add results from the additional pretrained text encoder experiments to an appendix.

Purely stylistic, not essential: capitalise words in all tables, e.g. in table 1: Random, GroupViT, Baseline, Ours, Subj, Verb, Object, Overall

Figures 2 and 3: increase font size

Go through references carefully to ensure that:
-  venue names are properly capitalised (example: Nicolas Carion, Francisco Massa, Gabriel Synnaeve, Nicolas Usunier, Alexander Kirillov, and Sergey Zagoruyko. End-to-end object detection with transformers. In European conference on computer vision,pp. 213–229. Springer, 2020.)
- published versions (not arXiv) are cited when available (example: Samuel R Bowman, Luke Vilnis, Oriol Vinyals, Andrew M Dai, Rafal Jozefowicz, and Samy Bengio. Gener-
ating sentences from a continuous space. arXiv preprint arXiv:1511.06349, 2015. was published at CoNLL 2016: https://aclanthology.org/K16-1002/)
- venue names are consistent (example: Aniket Didolkar, Andrii Zadaianchuk, Rabiul Awal, Maximilian Seitzer, Efstratios Gavves, and Aishwarya Agrawal. Ctrl-o: language-controllable object-centric visual representation learning. In Proceedings of the
Computer Vision and Pattern Recognition Conference, pp. 29523–29533, 2025. vs: Soravit Changpinyo, Piyush Sharma, Nan Ding, and Radu Soricut. Conceptual 12M: Pushing web-scale
image-text pre-training to recognize long-tail visual concepts. In CVPR, 2021.)

**Audience:**

Yes

**Audience Explanation:**

All reviewers concur that the topic is of interest to TMLR. The paper explores improvements to CLIP that focus on the text encoder whereas most work focuses on the vision encoder. CLIP forms a backbone to many other systems so contines to be an important subject of research. Reviewer Zr52 states: "The findings of the paper, whether in favour of such representations or not, will be useful to the community and would provide valuable insights for people working in this area." I urge the authors to consider this point when making the requested changes.

**Claims And Evidence:**

No

**Claims Explanation:**

The paper makes two main claims: firstly that grouping tokens gives VLMs a better fine-grained understanding of vision and language, and secondly that the token groups discovered correspond to meaningful units.

Reviewer Fdbq asked for a range of additional experimentation, including using a pretrained text encoder in the ablation of reconstruction loss, and finetuning the vision encoder. They also ask for a quantitative analysis of the existence of effective textual units with statistical significance.

Reviewer Zr52 was not fully convinced of the claim that grouping tokens gives VLMs a better fine-grained understanding. This reviewer asked for more extensive baselines, and for experiments to be run over multiple seeds to see whether the improvements over baseline are statistically significant. Furthermore, if they are not, the claims regarding improvement should be tempered.

Reviewer USZ4 made the critique that the presented model is not particularly novel; however, as this is not a criterion for TMLR, this is not taken into account. Furthermore, the authors argue effectively for the novelty of their approach. The reviewer asks for more downstream tasks and baselines, and for addition to the related work. The authors have responded effectively to this and added to the related work in the paper.

The authors have made effective responses to the requests for additional experimentation from Reviewer Fdbq, and they have also run experiments over multiple seeds for SVO-Probes and FOIL-COCO, reporting the mean and standard deviation. They make an effective response to the request for more baselines. However, what is still lacking are the following *essential* changes:

**Statistical significance reporting for ablations in table 4**. In this table, some of the improvements are very small (e.g. tIoU with (76.42) and without (76.18) contrastive loss), and significance should be calculated for this. One method that does not involve rerunning experiments is to use bootstrapping (https://en.wikipedia.org/wiki/Bootstrapping_(statistics)). This allows you to calculate significance quickly and easily.

**A reconsideration of the strength of the claims**. In terms of improvement of understanding, although the overall improvement on SVO Probes is significant (Table 1 and section 4.2.1), the improvements are really quite small: 0.6 percentage points. Statements like "but our multi-vector representation does a *much better* job of representing objects" (my emph) are too strong. Similarly, for FOIL-COCO (Table 2, Sec 4.2.2) the phrase " our model demonstrates a *remarkably good* performance, outperforming the transformer model in mean performance (paired t-test, p = 0.17)." is too strong, given an improvement of 0.69 percentage points. I also note that p=0.17 is high, and would not usually be considered a significant improvement.

While I think the work done in the paper is useful, the claims themselves must be tempered, as requested by reviewer Zr52, and significance for Table 4 should be calculated, as requested by reviewer Fdbq. However, rewriting the claims made and running bootstraps for the results in Table 4 can, I believe, be viewed as minor revisions.